ecology/complexity/computational biology

occupancy patterns, ant exploration, spin-glass

**Author for correspondence:**
Javier Cristín
e-mail: javier.cristin@uab.cat

# Occupancy patterns in superorganisms: a spin-glass approach to ant exploration

Javier Cristín[1], Frederic Bartumeus[2,3,4], Vicenç Méndez[1] and Daniel Campos[1]

[1]Grup de Física Estadística, Departament de Física, Facultat de Ciències),
Universitat Autònoma de Barcelona, 08193 Bellaterra, Barcelona, Spain
[2]Centre d'Estudis Avançats de Blanes (CEAB-CSIC), Carrer Cala Sant Francesc 14 17300 Girona, Spain
[3]ICREA, Institut Català de Recerca i Estudis Avançats, 08010 Barcelona, Spain
[4]CREAF, Centre de Recerca Ecològica i Aplicacions Forestals, 08193 Barcelona, Spain

JC, 0000-0003-2091-0346; FB, 0000-0001-6908-3797

Emergence of collective, as well as superorganism-like, behaviour in biological populations requires the existence of rules of communication, either direct or indirect, between organisms. Because reaching an understanding of such rules at the individual level can be often difficult, approaches carried out at higher, or effective, levels of description can represent a useful alternative. In the present work, we show how a spin-glass approach characteristic of statistical physics can be used as a tool to characterize the properties of the spatial occupancy patterns of a biological population. We exploit the presence of pairwise interactions in spin-glass models for detecting correlations between occupancies at different sites in the media. Such correlations, we claim, represent a proxy to the existence of planned and/or social strategies in the spatial organization of the population. Our spin-glass approach does not only identify those correlations but produces a statistical *replica* of the system (at the level of occupancy patterns) that can be subsequently used for testing alternative conditions/hypothesis. Here, this methodology is presented and illustrated for a particular case of study: we analyse occupancy patterns of *Aphaenogaster senilis* ants during foraging through a simplified environment consisting of a discrete (tree-like) artificial lattice. Our spin-glass approach consistently reproduces the experimental occupancy patterns across time, and besides, an intuitive biological interpretation of the parameters is attainable. Likewise, we prove that pairwise correlations are important for reproducing these dynamics by showing how a null model, where such correlations are neglected, would perform much worse; this provides a solid evidence to the existence of superorganism-like strategies in the colony.

# 1. Introduction

Collective behaviour is an adaptive widespread mechanism in biology present at many different organization levels. Phenomena such as bacteria self-organization [1,2], bird synchronization [3,4], movement in groups [5] and human social networks [6] are paradigmatic examples where emerging macroscopic properties come out of microscopic interactions.

Movement patterns, in particular, represent a common and relevant proxy to quantify classes of collective behaviour. Despite the great advances done in the field over the last decades, understanding how individual interactions yield collective patterns of foraging, home range dynamics, site-fidelity prevalences, or trail formation remains a productive and open challenge [7–10]. This is particularly relevant when trying to assess the idea of a superorganism, that is, a much more coordinated and coherent type of collective motion performed by cognitively limited interactive agents that ensures the survival as a whole (the superorganism) but not necessarily at the individual level [11,12]. Hence, the interactions between individuals should satisfy selection pressures and adaptability at the collective rather than the individual level. In such cases, it is crucial to characterize the system as a whole, with macroscopic parameters that can quantify the overall metabolic rate (energy) and the dynamics of the interactions at the superorganism level. Variation in these parameters may help us to understand the constraints and plasticity of self-organized behaviour under different selective pressures.

Nowadays, current high-resolution tracking and visualization technologies allow us to quantify much better behaviour [13], and more specifically the complexity of behavioural interactions among agents, offering us the opportunity to account for the challenges posed by biology when characterizing multi-agent systems. So far, many studies of collective motion have focused on informed-naive or leader-follower relative positions and behavioural relationships [14]. As a result, an increasing amount of experimental work is disentangling pairwise interaction rules for particularly simple cases of group dynamics, as is the case of fish schooling [15,16] or locust swarms [17,18].

Within this context, statistical physics represents a very convenient framework that provides powerful toolsets to ecologists to understand the theoretical foundations of emerging phenomena [19,20]. Specifically, statistical physics focus on revealing the *minimal* rules or models that are required to yield specific macroscopic properties out of interactions among individual entities. Ecologists can compare such minimal requirements to real interactions within the population to provide more detailed species- or case-specific analysis.

One additional advantage of statistical physics, given its abstract nature, is that it can be easily adapted for application at many different scales or levels. This is specially convenient when dealing with complex systems, which are known to possess relevant information at different scales. For example, spike patterns in neuronal systems can be studied either at the level of single neurons or rather meaningful regions can be grouped together or clustered to simplify the analysis [21,22]. Then, if microscopic details of interaction between the individual units (e.g. organisms) are not clear or available, alternative strategies can consist of studying patterns at other (still meaningful) levels.

We propose in the present work to apply such strategy for the study of collective movement patterns in populations. Despite the technological and methodological advances stated earlier, characterization of interactions between organisms at a fine level of detail, for most ecological situations and/or species, is still not possible. So one can still wonder if alternative levels of information exist that can be used and explored instead to study such collective effects. Our hypothesis here is that collective and social organization within groups (and superorganisms, in particular) should reflect in a non-trivial spatial use and, as a result, in complex space occupancy patterns. Then using spatial sites (not organisms) as the main units of interest and applying statistical physics tools at that level would represent a meaningful way to elucidate the existence of such complexity.

The main objective of our work thus is to present a new methodology for studying these occupancy patterns. As has been already done for the study of other complex systems as in neuronal activity patterns [21,22] or in social networks [23], we use for this purpose a spin-glass approach (so named because it was originally proposed within statistical mechanics as a model for understanding disordered glasses consisting of elementary spins/atoms). This is known to be the minimal model (from the point of view of entropy maximization principles), which can reproduce the dynamics of a system when it is to be described just in terms of pairwise correlations between their fundamental units [24]. As we will show, we use the existence of those correlations (in our case, correlations between occupancies at different spatial sites) as a proxy to identify the existence of socially planned (that is, superorganism-like) strategies in the spatial patterns.

To facilitate understanding of our methodology, we will present in the following its theoretical grounds (§§2 and 3) and will also show how to apply it for a particular situation of interest (§§4–7).

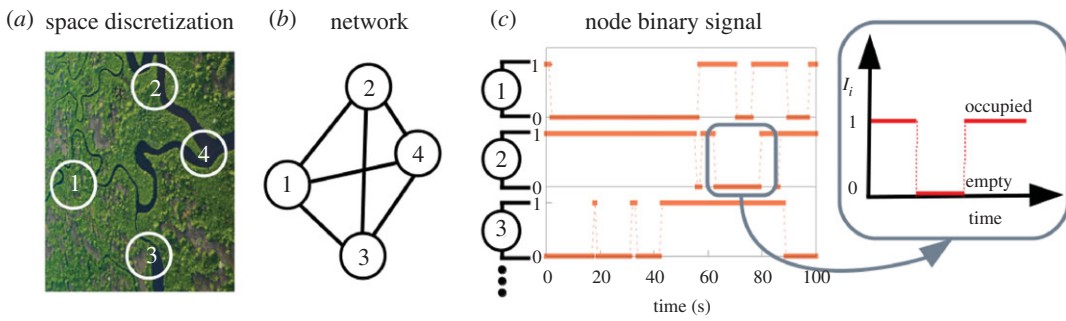

**Figure 1.** Illustrative scheme of our approach to collective space use. (*a*) The biologically relevant regions of the space are considered as (*b*) nodes of a network structure. (*c*) The presence or absence of individuals at node *i* for a given time *t* is then understood as a binary signal $I_i$ (*t*) representing occupancy of that region (with $I_i$ (*t*) = 1 if occupied, and $I_i$ (*t*) = 0 if empty).

For the latter, the occupancy signals of a colony of ants in an artificial arena (specially suited to fit our approach) will be used. Note that social insects represent an optimal model to test our approach as they have been claimed to represent a paradigmatic case of superorganism behaviour [25,26]. Task allocation in social insects [27–29] is a good example of how adaptive superorganism-like strategies emerge from individual interactions within the group. Ants, in particular, use simple communication rules (pheromone deposition and/or antennae contacts, basically) to adjust (i) recruitment efforts [30–32], (ii) the structure and intensity of foraging trails as a function of the resources available [33,34], and/or (iii) the amount of cooperative transport [35]. The analysis of our experiment, then, is focused on providing evidences of superorganism-like behaviour in ants from the comparison between the spin-glass approach and the experimental occupancy patterns by the colony. Other potential uses of our approach, however, are discussed in Discussion section for the sake of completeness.

## 2. Spatial networks and collective occupancy patterns

Because the goal of our work is to present a methodology to analyse how groups explore and exploit collectively different regions of space (e.g. food spots, habitats), it is first convenient to interpret each of those regions as interacting units embedded within a spatial network structure. Networks have been widely used in biology [36,37] at many different levels before: prey–predator interactions [38], extinction dynamics in food webs [39] or hierarchical organization in social animals [40,41] are just a few examples. For these, the network nodes can be identified as the different species inside the same ecosystem (this can be useful to study species or population survival and functioning within ecosystems) or as the individuals within a given group (when the aim is to understand connections in a hierarchical group). Within our approach, however, nodes represent biologically meaningful regions or sites in space. In figure 1*a,b*, we illustrate the idea that even for the case of relatively homogeneous landscapes, the environment can be always properly discretized or partitioned according to some case-specific criteria. We stress that such kind of discretization is convenient in many biological studies and can be exploited at different levels. This happens for example when using the concept of *patches* in fragmented habitats [42], but also in metapopulation theory (where a population is fragmented in several groups) [43,44], in the context of polydomy (where a colony is established across multiple nest sites) [45], or in theoretical approaches as cellular automatas [46], to name a few.

To characterize occupancy at each node of the spatial network, we will assume that a time-discrete occupancy signal $\mathbf{I}(t) = \{I_1 (t), I_2 (t), I_3 (t), \ldots, I_N (t)\}$ is available experimentally. Here, $I_i$ (*t*) (with *i* = 1, 2, …, *N*) will be taken for simplicity as a binary variable that tells us simply if the node *i* is occupied ($I_i(t) = 1$) at time step *t* or not ($I_i(t) = 0$), with *N* the total number of nodes in the spatial network. We note that more complicated versions of the approach could be proposed by relaxing this two-state (empty/occupied) hypothesis, while we prefer to focus on this case here to keep the notation and analysis simple.

The overall signal $\mathbf{I}(t)$ (figure 1*c*) then carries the information about the occupancy patterns for a group of organisms. If the individuals within that group behave independently, then occupancy patterns at each site, or node, will be independent of each other. On the contrary, one should expect that in a superorganism-like system (that is, for strongly organized groups exploring space according to some interaction rules and a global strategy) correlations between sites will emerge. Accordingly, a

minimal model able to capture such possible correlations would be convenient. Furthermore, for predictive purposes and subsequent testing, such an approach should be able not only just to identify the statistical properties of occupancy signals $\mathbf{I}(t)$ but also to: (i) generate new artificial realizations of those occupancy patterns; and (ii) be flexible and meaningful enough to allow for introducing variations in the model as a way to inquire or predict its behaviour under different conditions. In the following, we illustrate how spin-glass approaches can fulfil these requirements.

## 3. Spin-glass approach for occupancy dynamics

Spin-glass models can be seen as a generalized version of the ubiquitous Ising model from statistical physics. While the Ising model was originally aimed at deriving macroscopic properties of magnetic systems from elementary pairwise interactions, it has been subsequently extended in many different ways. In particular, introducing randomly distributed intensities for pairwise interactions has become a paradigm for frustrated systems, spin glasses being a recurrent example. This idea has also pervaded many other areas of research, including biology, up to the point that recent works [47] sustain that the physical foundations for the origin of complexity in biology can be outlined within that framework of frustrated systems and spin glasses.

Following the usual formulation of Ising-like systems, we will consider $N$ interacting units or *spins* (which in our context correspond to the sites within the spatial network structure) whose individual dynamics is described by a binary signal $s_i(t)$, with $s_i(t) = \pm 1$, so the overall state $\mathbf{S}$ of the system is characterized by a specific realization of each, and it is $\mathbf{S} = \{s_1, \dots, s_i, \dots, s_N\}$. The spins are assumed to interact through pairwise (both short and long range) interactions with intensities $J_{ij}$ and are subject to external fields of intensity $h_i$, such that the resulting Hamiltonian for the spin glass reads

$$H(\mathbf{S}) = -\sum_{i=1}^{N} h_i s_i - \sum_{i<j} J_{ij} s_i s_j, \tag{3.1}$$

where the second sum extends over all possible pairs of spins in the system.

As stated elsewhere (see, for example [24] for a comprehensive discussion), this Hamiltonian also admits an interpretation from information-theoretic grounds. In particular, it corresponds to the energy functional that minimizes redundancy (or, equivalently, maximizes informational entropy) for modelling the system, provided that the only information available are the time averages of the spins (it is, $\langle s_i(t) \rangle$) and their pairwise time correlations $\langle s_i(t) s_j(t) \rangle$. For this reason, the spin-glass model is typically employed for assessing whether complex systems can be possibly characterized or not only through pairwise correlations. This idea has been already used, for example, to study the spike dynamics of different kinds of neuron groups [21]. The spike/silent periods characteristic of these systems can be mapped into a binary signal representing the activity of each neuron, such that activity patterns of a group of neurons then provide information about their functional response and connection patterns. Furthermore, in recent years, there has been a growing interest in exploring the applicability of the spin-glass approach to different biological and social systems, including social human and animal connections [22,23] or financial markets [48,49]. The use of those models to study spatial dynamics in biology, however, represents a novelty of the present work as far as we know.

Without losing generality (and just for the sake of consistency with previous works on spin-glass models [21–23]), in the following, we will explore the statistics of occupancies $I_i(t)$ (which take values 0 or 1) from that of spins $s_i(t)$ by means of the mapping $s_i(t) = 2I_i(t) - 1$. So, given the experimental signal of node occupancies, we can then immediately find the corresponding states $\mathbf{S}$ and their probability distribution $P_{\exp}(\mathbf{S})$.

Under equilibrium conditions, the spin-glass system is expected to satisfy a Boltzmann distribution, so

$$P(\mathbf{S}) = \frac{1}{Z} e^{-\beta H(\mathbf{S})} \tag{3.2}$$

must hold, where $Z$ represents a normalization factor and $\beta = 1$ will be used from now on for the sake of simplicity.

The *inverse Ising approach* [24] is the name given to the procedure that consists of determining the set of parameters $h_i$ and $J_{ij}$ in the spin-glass Hamiltonian (3.1) that provide the best fit between experimental signals and the equilibrium condition (3.2). Note that the model contains $N$ parameters of type $h_i$ and $N(N-1)$ parameters of type $J_{ij}$, for a total of $N^2$ parameters in the system. Then, as long as $N$ is relatively large, such fitting procedure becomes extremely complex and requires the use of machine

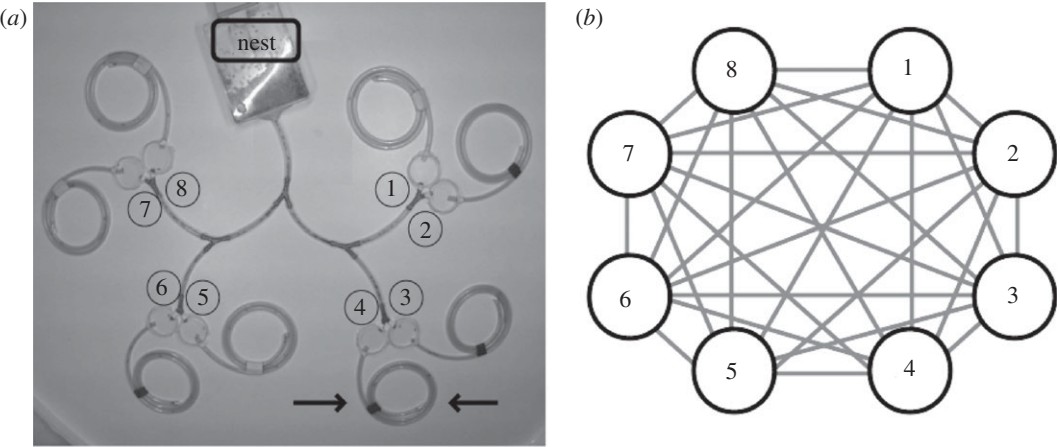

**Figure 2.** (*a*) Experimental set-up, labelling from 1 to 8 the nodes representing the endings of the tree structure. The black arrows indicate the plastic spirals where the food is laid. (*b*) Equivalent network representation of the experimental set-up, assuming every node can interact with any other.

learning algorithms. In our particular case, we will use a Boltzmann machine [50] for this purpose, which is a stochastic variation of the deterministic Hopfield neural networks [51,52]. We use the Boltzmann machine to determine values for $J_{ij}$ and $h_i$ such that, when introduced into the spin-glass model (equation (3.1)), generate a simulated distribution $P_{sim}(\mathbf{S})$, which minimizes the statistical divergence with the experimental one (further details of the implementation of the algorithm are given in appendix B).

We note that this procedure requires sampling the whole distribution of states and assessing their individual probabilities, so it is only computationally attainable when the number of nodes $N$ is relatively low, as is the case of the example we will present in the following section (for which $N = 8$). When the network is formed by a large number of nodes, however, computational costs grow very fast. Alternative methods, based on partial approximations to the expected distribution, are then required; the reader is addressed to previous studies [21,23,24] for a deeper discussion on this topic.

## 4. Experimental set-up and data analysis

To illustrate the application of the method presented in the previous section to a real situation, we carried out an experiment of ant foraging under laboratory conditions specifically designed to suit the spatial network approach in figure 1. We expect this experiment to provide a direct measure/evidence of superorganism-like behaviour in ants by a comparison between the spin-glass approach and the experimental occupancy signal exhibited by the colony (§5).

We put a colony of around 100 workers of *Aphaenogaster senilis* (previously collected from the Campus of the Universitat Autònoma de Barcelona) in a plastic nest together with several eggs and larvae to keep them stimulated for working. The eggs and larvae were not renovated, but they were left to grow and become adult during the experiment; while this introduced some non-stationarity in experimental conditions, we note that the duration of the whole experiment (25 days) was relatively short compared with the time required for the ants to mature to the adult stage, so we expect that this bias had minimum effects on our results. The colony nest was connected to a tree structure (figure 2) by a plastic tube finalizing in eight different Petri dishes of 4 cm diameter. Each of these dishes (nodes) is then connected to a dead-end spiraling plastic tube, which is used as the resource area where food for the ants is laid (see figure 2). While such a ramified structure does not represent a very natural environment for foraging, we stress that such configuration provides an ideal opportunity to explore how the ant colony distributes their exploration resources (scouts) in a discrete network environment. In particular, the eight Petri dishes in the structure (which have different topological distances between them) will represent from now on the eight 'nodes' (see figure 2*b*) of our network, in analogy with figure 1.

The implementation of the experiment consisted of daily trials of 90 min during 25 days (carried out in June and July 2018), which were recorded using a time-lapse camera at a frame rate of 1 Hz. During the 90 min of the trial, ants were allowed to explore the structure starting from the nest. Before each trial, the food was placed at some of the resource areas (dead-end spiraling tubes) departing from the eight nodes. Food consisted of single small mealworm pieces, so recruitment through pheromone was avoided

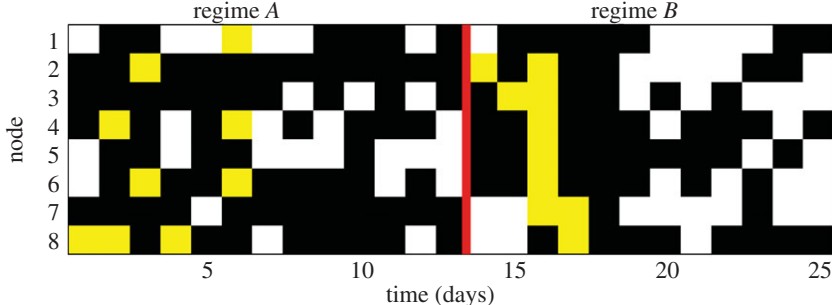

**Figure 3.** Food dynamics in the network nodes during the 25 days of experiment. Black cells correspond to nodes where food was not present at that day, white regions correspond to nodes which contained food and it was picked up by the ants, and yellow cells correspond to nodes where the food was present but it was not picked up by the ants during the trial. The red line separates the regimes $A$ and $B$ used in our analysis.

(according to previous studies on the recruitment rules of *A. senilis* [53]) and foraging was expected to rely on memory and landmarks solely. At the end of the 90 min trial, we removed the remaining food in case the ants had not collected it. The overall quantity of food received by the colony according to this procedure (about *three* pieces of mealworm per day, in average) was enough to nourish the colony while still keeping it active for foraging.

The resource areas containing food at each trial were decided according to probability rules (see figure 3). They were used as a control variable in order to evaluate the response of the colony to induced resource heterogeneity. So that, for days 1–13, half of the nodes had a high probability to contain food (50%), and the other half had a lower probability (25%), while in days 14–25, the probabilities were inverted (nodes with 50% probability turned to have 25%, and vice versa). Figure 3 shows the 25 day profile of the nodes where food was laid before the experiment, and this was the food collected by the ants.

We expected to observe changes in the occupancy patterns of the network as a result of variations in the resource availability on the nodes from food regime $A$ (days 1–13) to regime $B$ (days 14–25). Although it was not possible from the experimental design to assign directly the changes observed in space occupancy pattern to the change in the resource distribution (owing to a mixture of effects present in the system, including memory and/or habituation effects), we simply intended to check if variations in the external conditions driving foraging could reflect into a change in the collective occupancy patterns, which we could observe through our spin-glass approach. So, despite not controlling the exact biological forces that may govern the behaviour at both regimes ($A$ and $B$), we decided for convenience to analyse them separately to quantify their differences.

For extracting the experimental data, we used our own video analysis code (implemented in the open-source software *Scilab*) to determine the presence or the absence of ants in each one of the eight nodes at each frame (as stated earlier, we considered a node $i$ as occupied whenever one or more ants were detected at that node). After the processing of the videos, the complete dataset of 135 000 frames (which is available as the electronic supplementary material) was transformed into an occupancy binary signal $\mathbf{I}(t)$. So, every frame was characterized by a set of eight binary variables, for a total set of $2^8 = 256$ possible states where the system can be found.

We studied the daily profile of the average occupancy $\langle I \rangle$ (which corresponded to the occupancy at a given time averaged over the eight nodes) during the 90 min of experiment (i.e. 5400 s). From this we found that, as expected, the dynamics during food regimes $A$ and $B$ were significantly different (see figure 4$a$,$b$,and the corresponding caption). At the beginning of each daily trial, the ants were in the nest and it took some time before the nodes were visited for the first time. Then, the average occupancy during this transient period reads $\langle I \rangle = 0$. After that period, there was a tendency for $\langle I \rangle$ to grow until it reached saturation in 20 or 30 min.

If we split our analysis into those nodes where food had a higher (50%) or lower (25%) probability of being laid, we observe that for regime $A$ the ants were clearly visiting the nodes with more food with a higher frequency (figure 4$a$). However, for regime $B$, the ants performed a seemingly homogeneous exploration of the nodes independently of the presence of food (figure 4$b$). If we instead split our analysis into nodes that actually have food or not, we recover very similar qualitative differences between the regimes (see appendix A). So, the differences detected are robust and independent of whether we use present-day food or past averages as a criteria for splitting nodes.

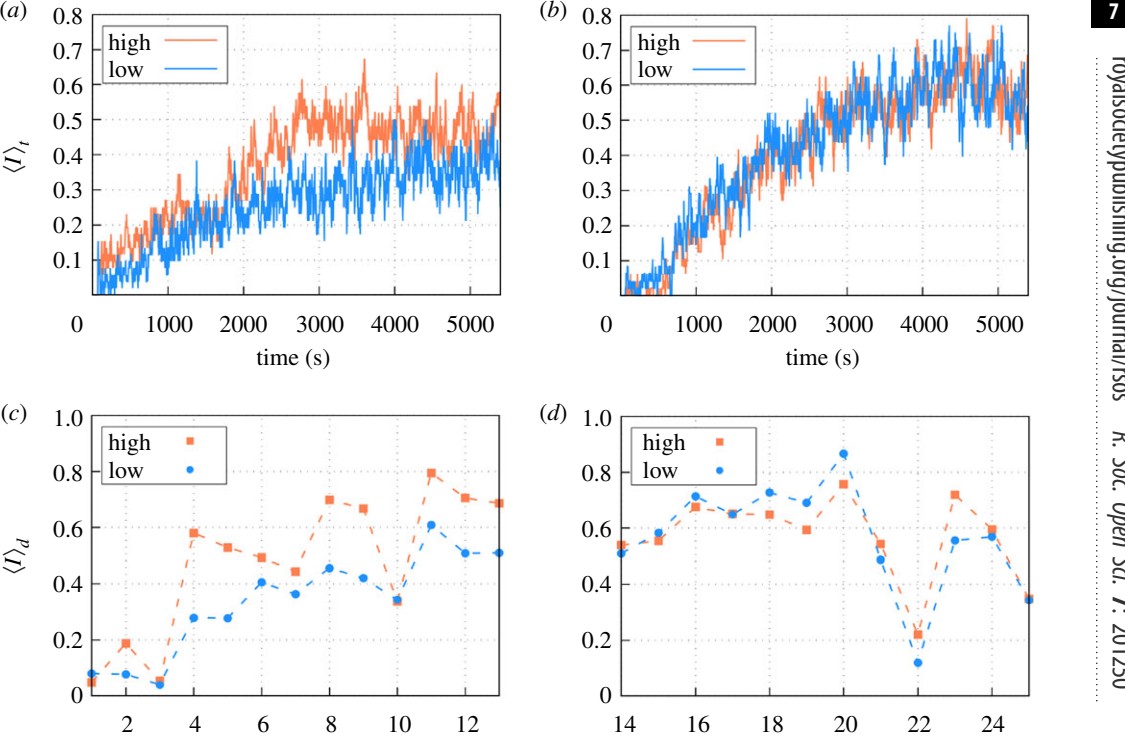

**Figure 4.** Averaged daily occupancy $\langle I \rangle_t$ for (a) regime $A$ ($\langle I \rangle_t^h = 0.48$; $\sigma_I^2 = 0.012$ and $\langle I \rangle_t^l = 0.34$; $\sigma_I^2 = 0.004$) and (b) regime $B$ ($\langle I \rangle_t^h = 0.57$; $\sigma_I^2 = 0.004$ and $\langle I \rangle_t^l = 0.57$; $\sigma_I^2 = 0.006$). Averaged occupancy split into nodes with high or low probability of containing food during (c) regime $A$ and (d) regime $B$. Differences between the two types of nodes are statistically significant ($T-$Student test) for the regime $A$ ($\langle I \rangle_d^h = 0.48$ and $\langle I \rangle_d^l = 0.34$; $|T_s| = 4.70 > T_{95\%} = 0.0002$) but not for the case $B$ ($\langle I \rangle_d^h = 0.57$ and $\langle I \rangle_d^l = 0.57$; $|T_s| = 0.11 < T_{95\%} = 0.46$).

For the sake of completeness, we also show the daily average of the occupancy throughout the 25 days of the experiments (figure 4c,d). From that we reach again very similar conclusions. For regime $A$, the nodes with larger quantities of food showed higher occupancy, while for regime $B$, the colony performed a more homogeneous exploration, independent of the quantity of food. In any case, this daily average shows that the occupancy is highly fluctuating throughout the days.

# 5. Spin-glass approach to ant occupancy patterns during foraging

We introduced the experimental dataset described in §4 into our machine learning algorithm for obtaining fitted values for the parameters $h_i$ and $J_{ij}$ through the inverse Ising approach mentioned in §3. Because in the experiments occupancy signals present a transient period until exploration of the space is possible for the ant colony (figure 4c,d), only the stationary part of the signal (period 2700s–5400 s) was used for the learning protocol. Learning procedures for regimes $A$ and $B$ are carried out separately to see how the differences observed in the space-use dynamics between both translate into our spin-glass approach. Because training is based on minimizing the difference between the probability distribution over the 256 possible states of the system and the probability distribution predicted from the spin-glass approach, before we carry out any subsequent analysis, we show that after the training process, the resulting spin-glass Hamiltonian is able to generate artificial occupancy signals (through Monte Carlo (MC) simulations) whose probability distribution is almost in perfect agreement with the experiments. Figure 5 provides this comparison and verifies that the agreement reached is excellent for both regimes $A$ and $B$.

## 5.1. Time dynamics comparison

Once the learning process has been satisfactorily completed, we run simulations to compare the properties of the occupancy signals produced by the model with the experimental ones. The agreement found at the level

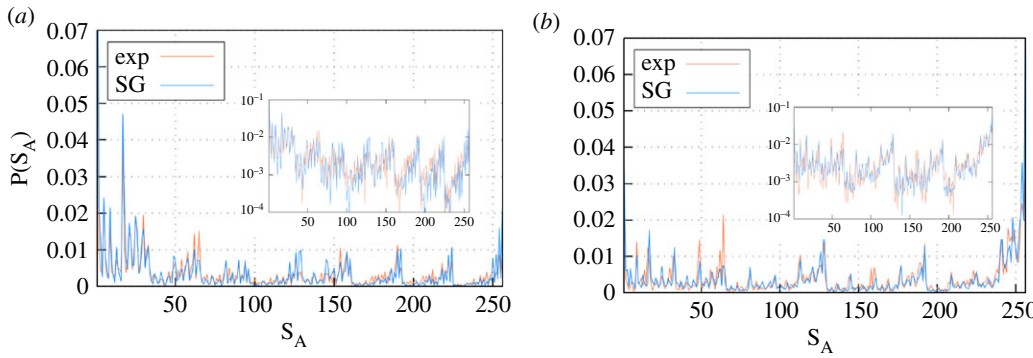

**Figure 5.** Probability distribution of states (the x-axis contains the $2^8 = 256$ possible states) for the experimental data and the spin-glass approach for (a) regime A and (b) regime B. The insets show the same probability distributions in the logarithmic scale. For further details on the goodness of the model, see appendix B.

of stationary probability distributions in figure 5 is to be expected by construction, but the spin-glass approach is not necessarily expected to reproduce the time dynamics of the experimental system beforehand. This is what we test in the following.

We first fix the initial state in the MC simulations as $\mathbf{I} = 0$ (or, equivalently, $\mathbf{S} = -1$), corresponding to all nodes empty, for reproducing the initial conditions of the experimental trials, and we study the behaviour of the corresponding average occupancy $\langle I \rangle$ during the 90 min of the trial. On average, the simulations recover almost completely the 90 min temporal signal of the experimental data for both regimes (figure 6a,b, and corresponding caption). Furthermore, we also find that the fluctuations around this average, measured through the corresponding standard deviation $\sigma$ among the different trials, are also very similar in all cases (figure 6c,d, and corresponding caption). The mean occupancy, when averaged over the whole time period of the trial for each node, also recovers the experimental pattern observed (figure 6e,f). Note that carrying out all these comparisons requires introducing a time scale in the spin-glass dynamics, because in principle, the evolution from $\mathbf{I} = 0$ to the stationary situation occurs through successive MC trials without any explicit time scale. For this, we used a rescaling in the signal from the MC simulations such that each simulation step is taken to correspond to 50 s (see appendix B for a further justification).

To expand on this idea, we also measure the distribution of persistence times at single nodes, defined as the amount of time one node stays in the same state (either occupied or empty) before switching to the other. When we look at the corresponding distribution of persistence times $P(\tau)$ experimentally for the ants, we find it follows a non-trivial behaviour which corresponds to an intermediate decay between exponential and power-law functions (figure 7). Although it is not possible to derive an analytical expression for that distribution, we observe that the spin-glass approach yields again a very good agreement to the experimental data, both for regimes A (figure 7a) and B (figure 7b). Altogether, these results prove that the spin-glass approach is able to reproduce dynamical properties, as well as the stationary statistics, in the experimental conditions used.

# 6. Biological interpretation of the model parameters

In some of the previous contexts where spin-glass approaches have been employed (including biology), the parameters $h_i$ and $J_{ij}$ of the characteristic Hamiltonian did not necessarily have a straightforward experimental interpretation because *spins* or units lived in an abstract phase space and could only be treated then as effective parameters. On the contrary, for the case of our experiment, we find that a relationship between the experimental data and these parameters becomes straightforward. The external field $h_i$, for instance, should determine the propensity that organisms have to occupy the $i$ node (i.e. the attraction towards node $i$) at an average level (equation (3.1)). When $h_i$ is large (small), provided that the other parameters remain unchanged, that node should have a tendency to be on average more (less) time occupied. The external field parameter $h_i$ should thus correlate with the average occupancy $\langle I_i \rangle$ of each node; this is confirmed by comparing the values of both magnitudes in both regimes considered (figure 8a).

According to the previous result, the parameters $h_i$ capture the average node occupancy; this should be good enough as long as one is not concerned about collective effects in the system. The role of the

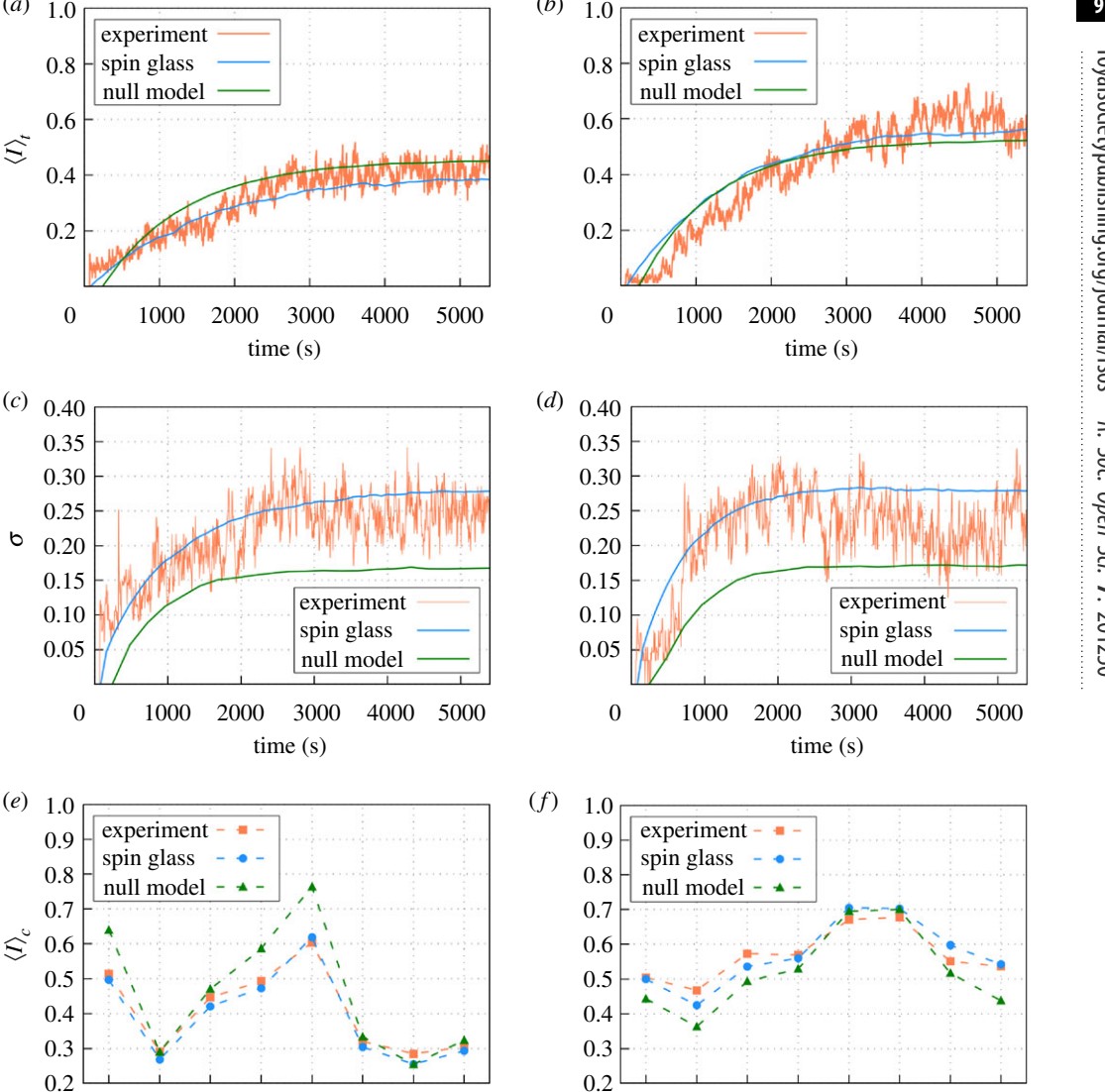

**Figure 6.** Averaged daily occupancy dynamics $\langle I \rangle_t$ for the spin-glass approach (with and without pairwise interactions) and experimental data during (a) regime A and (b) regime B. The corresponding standard deviations are also shown for (c) regime A and (d) regime B. Averaged occupancy in the stationary regime $\langle I \rangle_c$ for the spin-glass approach and experimental data during (e) regime A and (f) regime B. For more information of the relationship between the MC step and the real time, see appendix B. The deviation of $\langle I \rangle_c$ obtained from the models to the experimental one is computed from the variance of the departures from one to the other: (e) we find ($\sigma_{SG}^2 = 0.0004$ and $\sigma_{J=0}^2 = 0.0065$), and (f) ($\sigma_{SG}^2 = 0.0011$ and $\sigma_{J=0}^2 = 0.0025$). In both regimes, the spin-glass model shows smaller departures from the experiment than the null model. In addition, an analysis of variance (ANOVA) test with $\alpha = 0.05$ for null differences with the experimental data leads to $p$-value $= 0.80$ (spin-glass) and $p$-value $= 0.54$ (null model) for regime A, and $p$-value $= 0.96$ (spin-glass) and $p$-value $= 0.37$ (null model) for regime B.

interaction parameters $J_{ij}$, however, is extremely relevant because they capture spatial correlations at a collective level (in particular, how likely is it that two separate nodes/spots are simultaneously occupied by the colony). To confirm this intuition, we compare the $J_{ij}$ parameters, obtained from the learning process, with pairwise correlations of the experimental occupancy signal $\langle I_i I_j \rangle \equiv C_{ij}$ (figure 8b). We again find a high correlation between both magnitudes, so confirming our intuitive interpretation of the $J_{ij}$ parameters.

The alignment between the model parameters $h_i$ and $J_{ij}$, and the observables $\langle I_i \rangle$ and $C_{ij}$ are still found if computing average values separately for the nodes with high and low resource probability (figure 8).

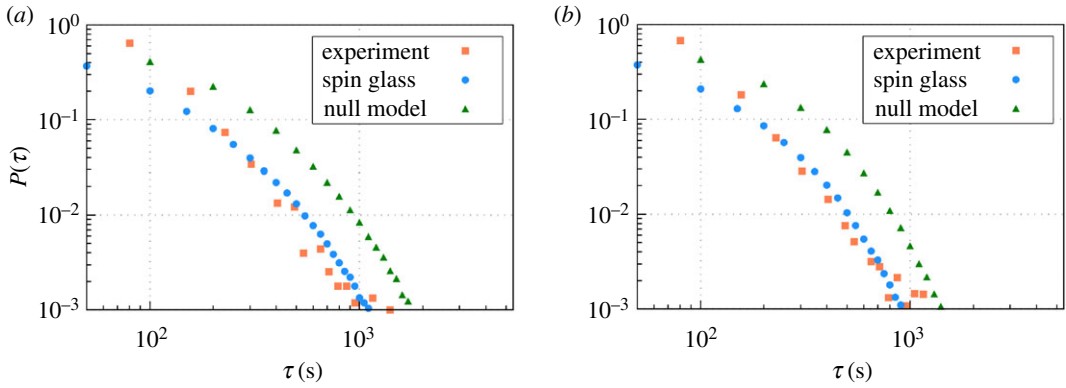

**Figure 7.** Probability distribution of switching times $P(\tau)$ during (a) regime A and (b) regime B. An ANOVA test with $\alpha = 0.05$ for null differences with the experimental data leads to p-value = 0.86 (spin-glass) and p-value = 0.04 (null model) for regime A, and p-value = 0.82 (spin-glass) and p-value = 0.09 (null model) for regime B.

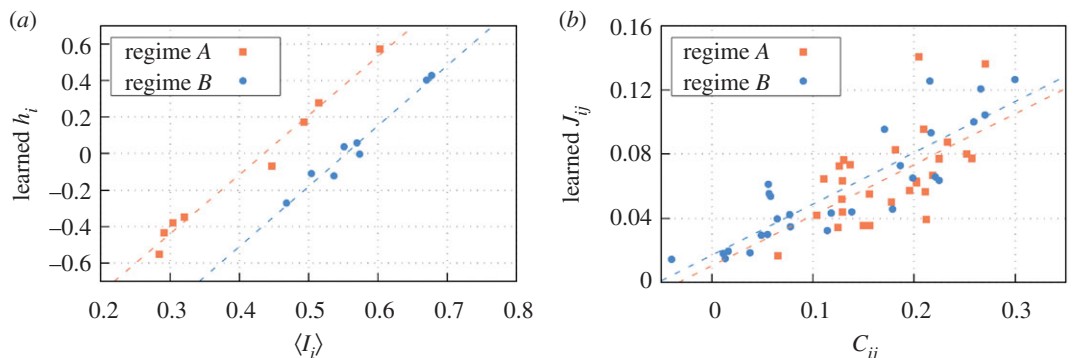

**Figure 8.** (a) Averaged experimental occupancy for each node $\langle I_i \rangle$ in comparison to the learned external field $h_i$. The correlation coefficients are $C = 0.991$ for the regime A dataset (orange) and $C = 0.982$ for the regime B dataset (blue). (b) Experimental correlation between the nodes i and j, $C_{ij}$, in comparison to the learned pairwise interaction $J_{ij}$. The correlation coefficients are $C = 0.597$ for the regime A dataset (orange), and $C = 0.876$ for the regime B dataset days (blue).

Of note, the average occupancy per node $\langle I_i \rangle$ and the spatial correlations $C_{ij}$, as well as the corresponding model parameters $h_i$ and $J_{ij}$, are statistically different in nodes with high and low resources (food) only for the case of regime A, but it does not happen for regime B. This suggests (as explained in §4) either an habituation of the colony to the experimental set-up or a memory-delayed process for the colony to adjust the shift in high/low resourced nodes from regime A to regime B. Whatever the underlying biological process, in regime B, the ant colony shows a more homogeneous activity, with no differences in attraction and correlation patterns based on the amount of resources in the nodes (figure 9), something that our spin-glass approach can adequately capture.

As a side note, the exact relationship between $\langle I_i \rangle$ and $h_i$ is actually influenced by the presence of the correlations $J_{ij}$, too. In regime A, for instance, we observed lower occupancy intensities $\langle I_i \rangle$ compared to regime B, and larger differences between high and low resourced nodes (figure 9). In such contexts of low occupancy, correlations $J_{ij}$ tend to keep nodes unoccupied, so to obtain certain levels of occupancy in a node i, higher attraction forces (i.e. $\langle h_i \rangle$) are necessary. This phenomena explains the higher level-off of regime A compared to regime B in the regression of figure 8a. In regime B, occupancy intensities are higher in general (figure 9a), and $J_{ij}$ then tends to keep the activity high in all nodes. Hence, to get certain level of activity in a given node i, a weak $h_i$ would be enough (level-off of the regression in regime B is smaller, see figure 8a). For example, an occupancy $\langle I \rangle = 0.5$ in regime A would require $h_i \approx 0.2$, whereas for regime B external fields can be much smaller, e.g. $h_i \approx -0.2$ (figure 8a).

Finally, one can also wonder about the effect that real topological distance between nodes may have on the $J_{ij}$ values, and whether there is an inverse correlation between both (so distant nodes show weaker interactions). Unfortunately, our results show that this is not the case in general. Because pairwise interactions can depend on several factors, distance by itself is not always a good predictor for them.

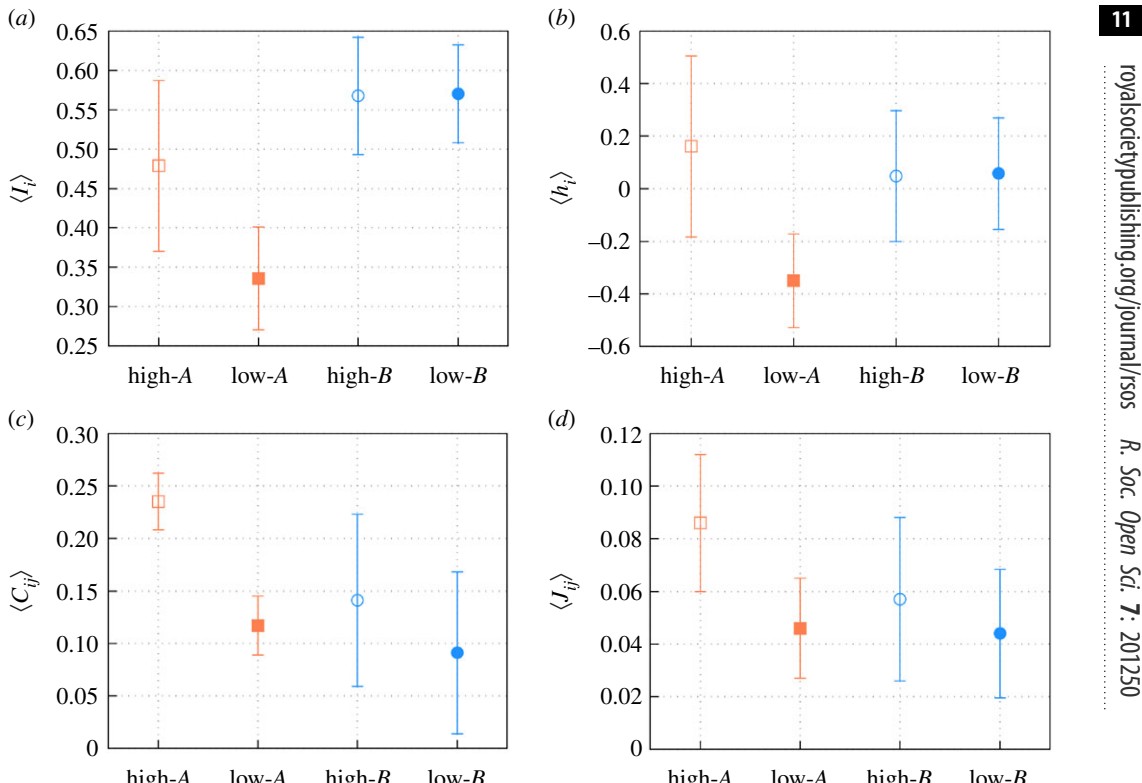

**Figure 9.** (a) Averaged experimental occupancy $\langle I_i \rangle$ for the different sets of nodes and experimental regimes $A$ ($T -$ Student test: $|T_s| = 2.21 > T_{95\%} = 0.06$) and $B$ ($|T_s| = 0.23 < T_{95\%} = 0.41$). (b) Averaged field $\langle h_i \rangle$ for the different sets of nodes and experimental regimes $A$ ($|T_s| = 2.82 > T_{95\%} = 0.03$) and $B$ ($|T_s| = 0.15 < T_{95\%} = 0.44$). (c) Averaged pairwise correlations $\langle C_{ij} \rangle$ between the nodes of the same set in the experimental regimes $A$ ($|T_s| = 8.71 > T_{95\%} = 0.00001$) and $B$ ($|T_s| = 6.90 > T_{95\%} = 0.00001$). (d) Averaged pairwise interaction $\langle J_{ij} \rangle$ between the nodes of the same set in the experimental regimes $A$ ($|T_s| = 3.83 > T_{95\%} = 0.001$) and $B$ ($|T_s| = 2.18 > T_{95\%} = 0.03$). High corresponds to the set of nodes with high probability of food presence, while low corresponds to the set of nodes with low probability. $A$ corresponds to the experimental regimes from days 1 to 13 of the experiment, while $B$ corresponds to the days 14 to 25.

In particular, for regime $A$, such a relation holds well, but for regime $B$, this breaks down completely (see appendix C for details).

# 7. Relevance of pairwise correlations in the occupancy patterns

We come back finally to the initial idea that led us to introduce our spin-glass approach. Our main objective is to check whether pairwise correlations between occupancies at different nodes are required to describe the overall occupancy dynamics. If so, then such correlations would indicate that some kind of global strategy is being used by the colony for exploring the arena and for distributing their foraging resources. On the contrary, if ants acted as individual foragers, then such correlations would be almost negligible or they would not have any significant effect on the occupancy pattern observed.

The existence of non-zero correlations $C_{ij}$ between nodes (figure 8) tells us that some collective effects emerge in ant colony space use, but one may still wonder whether these correlations are key to reproduce the experimental data. To assess that idea, we compare the results obtained from our spin-glass approach to a null model in which all pairwise interactions $J_{ij}$ are set to zero. After imposing such conditions, our machine learning algorithm is run again and the values for $h_i$ are determined anew. Also, to make the comparison fair, we redefine the time step in the MC simulations for comparing the artificial occupancy signal (obtained from the null model) to the experimental one; by doing so, we find that when single time steps in the simulation correspond to 140 s, we obtain the best possible fit for the average global occupancy (figure 6a,b, green lines). However, even if such rescaling is used to force that the average occupancies of the null model fit the experimental ones, the other properties of the occupancy signal perform poorly in comparison to the general model. In particular, fluctuations in

the overall occupancies (figure 6c,d, green lines), as well as characteristic switching times (figure 7, green dots), depart very much from those observed in the experiment, contrary to what happens for the general model (see figure captions to find the statistical significance of these departures). In addition, it is evident that if $J_{ij} = 0$ is imposed in the null model, then experimental correlations $C_{ij}$ found experimentally cannot be reproduced (for the sake of completeness, we show this in appendix D). All this together proves that the statistical patterns exhibited by the colony cannot be properly explained with the null model, giving support to the superorganism-like hypothesis.

# 8. Discussion

Ant foraging represents a paradigmatic example of collective intelligence [25]. Nonetheless, measuring the amount of collective activity, cohesion, and plasticity employed by such superorganisms in different contexts requires adequate modelling at different levels of description, and some of them are still unexplored. Here, we show that spin-glass models, when applied at the level of occupancy patterns in discretized media, can be used to describe the dynamics of multi-agent biological systems. Of note, this approach disregards the real behavioural details of individual interactions, the latter transformed into effective pairwise (spin-like) interactions among discrete regions (nodes) of a discretized landscape. This huge simplification accounts for rather simple assessment of the extent and type of superorganism-like behaviour with the experimental data.

As we have shown, from these powerful mathematical frameworks [47] not only can we replicate the space occupancy patterns at the colony level but also we obtain simple macroscopic parameters with direct biological meaning. The biological interpretation of the parameters of the spin-glass Hamiltonian (h's and J's) correspond to the ant average attraction to a given spot and ant position correlations across space, respectively. More specific attraction (large h's) means more activity efforts (which may be owing to recruitment or others) and larger metabolic rates expended by the superorganism (the ant colony as a whole) on a given spatially mapped task. Conversely, the larger the spatiotemporal correlations, the more likely that individuals are sharing information to perform the task, and the more cohesive is the response of the colony (large J's). Also, having a biological interpretation for the parameters in the model offers the possibility to tune, or adjust, such parameters as a way to infer how the system could behave under different external conditions. To give some examples, we could hypothesize that having a smaller (larger) colony would result in smaller (larger) values for $\langle I \rangle$, and so we would expect the parameters $h_i$ to be modified accordingly. One could also predict that diminishing communication between individual ants would translate into smaller $J_{ij}$ values, and the corresponding occupancy patterns could be predicted accordingly. In short, our approach generates a *replica* of the system (in terms of occupancies) that we can use for testing the consequences of altering the experimental conditions. Actually, we plan to explore such ideas for ant colonies in a forthcoming publication, while this article has been focused on presenting the approach for a particularly simple case of study.

Regarding the experiment used here, one might speculate that the two regimes (A and B) observed in the colony represent different organization within the superorganism dynamics when external conditions are modified (although this idea would require extra work for confirmation). In regime A, the overall activity of the colony was lower, but clearly distinct behaviour appeared in nodes with high and low resource probabilities. Attractive forces to highly resourced nodes were larger compared to low resourced nodes. High resourced nodes also showed more amplified occupancy correlations among them than low resourced nodes. This suggests a good spatial mapping with resource dynamics and a tightly cohesive behaviour as expected for a superorganism. On the contrary, in regime B, the overall activity was much higher, and the attractive forces to both types of nodes (i.e. high and low resourced) were homogeneous, suggesting a 'relaxation' of the superorganism-like behaviour, which despite showing spatial coherence (i.e. positive spatial correlations) did not map resource patterns as tightly as in regime A. Given that our system was pretty small and all nodes were easily accessible, we did not find remarkable differences in foraging success (figure 3). Nonetheless, at larger spatiotemporal scales, such differences may result in measurable and relevant foraging success differences.

Finally, a direct extension of our analysis to experiments under more natural conditions is straightforward provided that a meaningful habitat partitioning is introduced. One could study, for example, the overall degree of motivation, the attraction associated to given spots and the amount of spatial organization among different tasks performed by a single colony or among different colonies or species. After an appropriate implementation of the learning algorithm, the model should consistently reproduce the stationary and dynamical properties of the occupancy patterns. This

remains to be tested, as well as whether the continuum from simpler to superorganism-like strategies could also be identified in larger and complex domains, closer to field conditions. These appealing ideas will be the basis of future research.

Data accessibility. Data is included as the electronic supplementary material.
Authors' contributions. J.C., F.B., V.M. and D.C. conceived and designed the study. D.C. carried out the experiment. J.C. performed the numerical simulations. J.C., F.B, V.M. and D.C. carried out the mathematical analysis. J.C., F.B, V.M. and D.C. wrote and reviewed the paper. All authors gave final approval for publication and agree to be held accountable for the work performed therein.
Competing interests. We declare we have no competing interests.
Funding. This research has been supported by the Spanish government through grants no. CGL2016-78156-C2-2-R (J.C, D.C and V.M), FIS2015-72434-EXP (J.C, D.C and V.M) and CGL2016-78156-C2-1-R (F.B.)
Acknowledgements. We thank Xavier Espadaler for help with the experimental work.

# Appendix A. Analysis of node separation by food resources

The analysis of occupancies in §4 has been done by separating the nodes in two different groups: those with high, or low probability of having food resources (either for regime $A$ or regime $B$). However, we have also studied a different grouping that is biologically meaningful: the actual (or not) presence of food. In figure 10, we show that different classification for the nodes produces qualitatively the same behaviour shown in figure 4. We believe the high/low probability node classification is a good mechanism, so we have followed it during this work, because it has the advantage that we can always average over the same number of nodes.

# Appendix B. Simulation details

As we mentioned in §3, a Boltzmann machine algorithm [50] is used for implementing the learning process in the spin-glass approach that leads to a determination of the fittest parameters $h_i$ and $J_{ij}$. This involves the use of a gradient descent method in combination with MC simulations. We start from a random set of parameters $h_i$ and $J_{ij}$. At each MC step, a shift in one parameter (randomly chosen among all the $J_{ij}$ and $h_i$) is proposed. After that, we run the MC simulations with the new set of parameters to generate a new simulated distribution of states, $P_{sim}(S)$. If the new distribution improves the fit to the experimental distribution $P_{exp}(S)$, the shift in the parameter is accepted. To quantify the goodness of the fit between the simulated and the experimental distribution, we use the Kullback–Leibler divergence [54] between $P_{sim}(S)$ and $P_{exp}(S)$. If the new simulated distribution of states (after the shift) minimizes the Kullback–Leibler divergence, the shift in the randomly chosen parameter is accepted. We have performed training simulations of $10^6$ shift trials. We have seen all the training processes exhibit a stabilized Kullback–Leibler divergence before the end of the simulation. The parameters $J_{ij}$ and $h_i$ at the last are chosen as the fittest parameters in the model.

One aspect to remark on is the definition of a timescale in the MC simulations and how it is related to real time. This is necessary to establish a comparison between the time evolution of the experimental results and the ones by the model (MC simulations). By using as a criteria that the relaxation of the system towards its stationary state in the simulations should fit the experimental curve (figure 6) when starting from the state $\mathbf{I} = 0$, we conclude that a proper rescaling of the MC results consists of imposing that one step in the MC simulation corresponds to 50 s. That value is obtained elsewhere in the article for rescaling the results from the spin-glass model. The same criterion has been also applied for the null model presented in §7, and in this case, the rescaling that provides the best fit in figure 6 corresponds to 140 s for each step of the MC simulation. Consequently, the equivalence between the real time and the rescaled time for simulations is determined just by a parameter that shifts the relaxation to the stationary state, but does not affect the stationary dynamics in the system.

# Appendix C. Analysis of $J_{ij}$ by distance

As we have stated in 7, the averaged values of the pairwise interaction for both nodes with high and low probability of food resources are different between regimes $A$ and $B$ (figure 9d). As a complementary study, we here analyse the value of the pairwise interactions between nodes by distance (figure 11) measured as the number of tree bifurcations between the nodes. For regime $A$, the highest $J_{ij}$ value

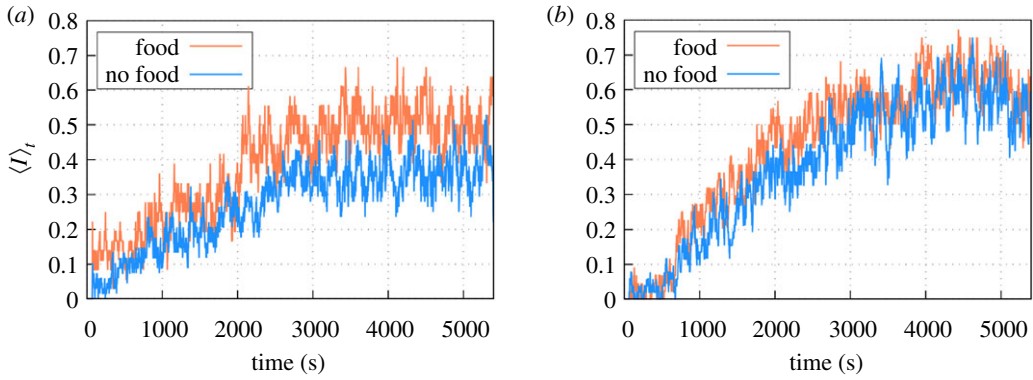

**Figure 10.** Averaged daily occupancy $\langle I \rangle_t$ for those nodes where food is present (red) or not (blue), for (a) regime A and (b) regime B. Comparison of stationary (2700–5400 s period) values leads to ($\langle I \rangle_t^f = 0.50$ and $\langle I \rangle_t^{nf} = 0.34$) for the case in (a), showing significant departures, while for case in (b) we find ($\langle I \rangle_t^f = 0.60$ and $\langle I \rangle_t^{nf} = 0.55$), much less difference is obtained.

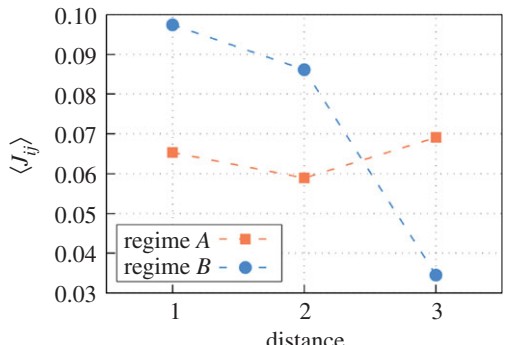

**Figure 11.** Averaged pairwise interactions ($J_{ij}$) between the nodes $i$ and $j$ by distance for regimes A and B.

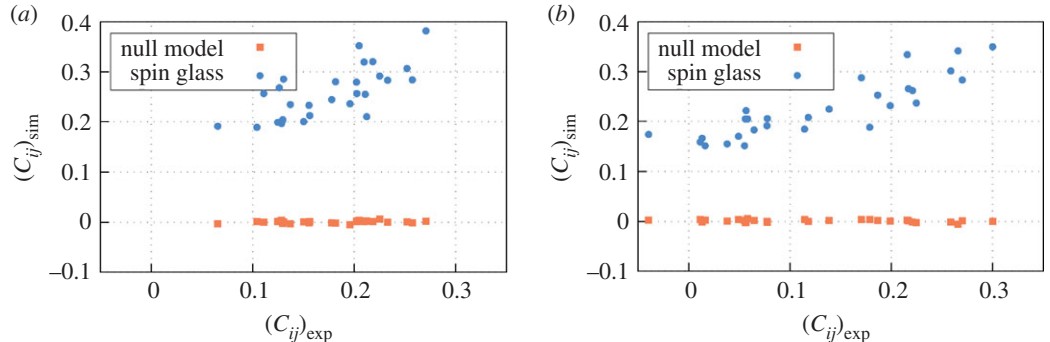

**Figure 12.** Experimental correlation between the nodes $i$ and $j$ ($C_{ij}$)$_{exp}$ in comparison to virtual correlations obtained from MC simulations ($C_{ij}$)$_{sim}$, for (a) regime A and (b) regime B.

corresponds to the largest distance (3 bifurcations). There is an interplay in the pairwise interaction between the distance and the food resources (see also figure 9d). In average, the distance does not have a strong effect in the $J_{ij}$. However, the behaviour of regime B is completely different. While the resources probability seems to have a smaller effect, the distance plays the key role in the value of the pairwise interactions (figure 11). According to this, simplified versions of the spin-glass model in which pairwise interactions ($J_{ij}$ values) were grouped in terms of distance would be uneffective to reproduce the patterns of all regimes.

## Appendix D. Correlations for the null model

As we have seen in §5, it is useful to compare the results from the general spin-glass approach to those of a null model where pairwise interactions are removed. To complete the discussion given in §7 regarding

this, we also provide in figure 12 additional evidence of how the model without interactions loses its capacity to reproduce the time correlations in the system (equation (D 1)). For the general model with interactions, the $(C_{ij})_{\exp}$ and $(C_{ij})_{sim}$ exhibit the same tendency. Instead, the correlations in the null model are all close to 0 owing to the removal of interactions between sites; so this model is not able to resemble the experimental correlations. This, again, proves the need for pairwise interactions for describing the system, so supporting the idea that spatial organization in the ant colony is guided by collective criteria/strategies:

$$C_{ij} = \frac{\sum_t (I_i(t) - \langle I_i \rangle_t)(I_j(t) - \langle I_j \rangle_t)}{\sqrt{\sum_t (I_i(t) - \langle I_i \rangle_t)^2 \sum_t (I_j(t) - \langle I_j \rangle_t)^2}}. \tag{D 1}$$

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
