## [Reviewer comments · Royal Society Open Science]

Review History

RSOS-201250.R0 (Original submission)

Review form: Reviewer 1 (Jacob Davidson)

Is the manuscript scientifically sound in its present form?

Yes

Are the interpretations and conclusions justified by the results?

Yes

Is the language acceptable?

Yes

Do you have any ethical concerns with this paper?

No

Have you any concerns about statistical analyses in this paper?

No

Recommendation?

Accept with minor revision (please list in comments)

Comments to the Author(s)

The motivation of the approach is much more clear in the revised version - in particular, that the paper is a Methods-focused paper, and that the experiment is done for the sake of being able to fit the model to it, and not to provide particular biological insight. With the addition of statistics, changes in wording, and changes in the comparison of the null model (also fitting the time constant), I think the comparisons are presented more clearly, and previous concerns have been addressed.

A minor point: whereas in other places statistical tests were added, in Fig 10 there is no test, so remove the wording 'significant'.

Decision letter (RSOS-201250.R0)

Dear Mr Cristín

On behalf of the Editors, we are pleased to inform you that your Manuscript RSOS-201250 "Occupancy patterns in superorganisms: a spin glass approach to ant exploration" has been accepted for publication in Royal Society Open Science subject to minor revision in accordance with the referees' reports. Please find the referees' comments along with any feedback from the Editors below my signature.

Please submit your revised manuscript and required files (see below) no later than 7 days from today's (ie 09-Nov-2020) date. Note: the ScholarOne system will 'lock' if submission of the revision is attempted 7 or more days after the deadline. If you do not think you will be able to meet this deadline please contact the editorial office immediately.

on behalf of the Associate Editor and Professor Pietro Cicuta (Subject Editor)
openscience@royalsociety.org

Editors' Comments to the Author(s):

Please accept our apologies for the delays experienced during peer-review of your paper. Unfortunately, it was unusually difficult to secure referees for your paper; mostly owing to the COVID-19 crisis. Nonetheless, we eventually managed to secure one of the original referees from your Interface paper, who is now largely happy with your manuscript. Please address the minor revisions suggested by the referee.

Thank you for your patience with this process. We look forward to a finalised version in due course.

Reviewer comments to Author:

Reviewer: 1

Comments to the Author(s)

The motivation of the approach is much more clear in the revised version - in particular, that the paper is a Methods-focused paper, and that the experiment is done for the sake of being able to fit the model to it, and not to provide particular biological insight. With the addition of statistics, changes in wording, and changes in the comparison of the null model (also fitting the time constant), I think the comparisons are presented more clearly, and previous concerns have been addressed.

A minor point: whereas in other places statistical tests were added, in Fig 10 there is no test, so remove the wording 'significant'.

===PREPARING YOUR MANUSCRIPT===

- one version identifying all the changes that have been made (for instance, in coloured highlight, in bold text, or tracked changes);
- a 'clean' version of the new manuscript that incorporates the changes made, but does not highlight them. This version will be used for typesetting.

===PREPARING YOUR REVISION IN SCHOLARONE===

Author's Response to Decision Letter for (RSOS-201250.R0)

See Appendix A.

Decision letter (RSOS-201250.R1)

Dear Mr Cristín,

It is a pleasure to accept your manuscript entitled "Occupancy patterns in superorganisms: a spin glass approach to ant exploration" in its current form for publication in Royal Society Open Science. The comments of the reviewer(s) who reviewed your manuscript are included at the foot of this letter.

on behalf of Dr Sean Rands (Associate Editor) and Pietro Cicuta (Subject Editor)
openscience@royalsociety.org

Associate Editor Comments to Author (Dr Sean Rands):
Associate Editor
Comments to the Author:
Thanks for being patient - I'm happy with this version.

Appendix A

Dear Editor,

First of all we are very grateful for the time you have invested in our manuscript, and we appreciate your effort

We have modified the minor changes the reviewers have stated. More precisely, we have removed the word “significant” in the Figure 10.

Yours sincerely,

The authors